# Participatory Live Coding and Learning-Centered Assessment in Programming for Data Science

**Sarah M Brown** [1]

## Abstract

Programming for Data Science is a programming intensive data science course. This paper discusses a revision of the course to center student learning. The revision effort centered the desired learning outcomes and resulted in a course that charted an explicit path toward achieving them for students. This paper summarizes the design overall and provides practical details about the instruction via participatory live coding and assessment with a competency based grading scheme.

## 1. Introduction

In this paper, we present an undergraduate course that teaches introductory data science through a programming intensive lens. As originally designed the course involved lectures using slides and prefilled Jupyter Notebooks followed by in class group work. In this format, the instructor provides the conceptual ideas for all material, foundational to advanced, and students figure out more practical details in group work and outside of class. Live coding flips the instructional model: the instructor provides core concept with their practical details and students build on the base knowledge to learn more advanced aspects independently. Active learning in this version was independent rather than in groups through formative assessment and following along coding. The changes in assessment were designed to create a more equitable and inclusive learning environment while maintaining high expectations for all students.

Section 2 describes the goals of the course through its context. Section 3 provides an overview of the design and how the pieces work together. The remainder of the paper goes into greater detail about how participatory live coding (Section 4) and learner-centric assessment (Section 5) worked in practice.

[1]Department of Computer Science and Statistics, University of Rhode Island, Kingston, Rhode Island, USA. Correspondence to: Sarah M Brown <brownsarahm@uri.edu>.

*Proceedings of the $2^{nd}$ Teaching in Machine Learning Workshop*, PMLR, 2021. Copyright 2021 by the author(s).

## 2. Course Context

Programming for Data Science is a required course for Data Science (DS) Majors and a popular elective for Computer Science (CS) Majors as it fulfills the requirement for a programming-intensive elective. The prerequisite is one programming course, but no statistics or math. CS majors take Computer Programming taught in C++, after Survey of Computer Science taught in Python. DS majors take Intro to Computer Programming taught in Python, which covers topics with less theoretical depth than the CS majors' course. Many students fulfill prerequisites at community colleges that teach in Java, so some students come to the class with no prior experience in Python. This course is a prerequisite to Machine Learning, which covers the implementation of machine learning algorithms and is also required for DS majors and popular among CS majors.

In this context, the role of this course is to give students a chance to deepen their programming skills and have an overview of data science so that they can succeed in understanding the algorithmic details of machine learning in a future course. To do this, communication about their work, data organization, examining results of machine learning models are essential as these skills will support students to focus on the machine learning algorithms in the subsequent course. For some students, this will be their only course exposure to machine learning concepts prior to graduation, so hands-on experience with a variety of models and a focus on careful selection of models is appealing. The course's five learning outcomes are: (*1*) Describe the process of data science, define each phase, and identify standard tools. (*2*) Access and combine data in multiple formats for analysis. (*3*) Perform exploratory data analyses including descriptive statistics and visualization. (*4*) Select models for data by applying and evaluating multiple models to a single dataset. (*5*) Communicate solutions to problems with data in common industry formats.

## 3. Design Overview

The course was developed using a reverse instructional design process, focused on guiding students to achieve the learning outcomes. To plan assessment, then instruction, the

learning outcomes were broken down into 15 component skills each of which was decomposed further to a 3 stage progression, shown in Table 1. The first level represents a basic understanding: the general terms and core concepts, typically at the understand level of Bloom's Taxonomy. The second level represents the ability to apply concepts with guidance as demonstrated in class, at the apply or analyze level of Bloom's Taxonomy. The third level is the ability to apply the general concepts beyond the scope demonstrated in class, operating at the evaluate or create levels of Bloom's Taxonomy. Each level of each skill is called an *achievement* and these served as the basis of grading.

The content of the course was sequenced to build skills early that would support later skills and activities were crafted to review prior topics. We used loading data as a lens to review of basic programming and reinforcement of the overview of Data Science. Next, we covered exploratory data analysis using pandas in order build understanding of what well structured data looks like and make the concepts of data science more concrete. Then we covered Data cleaning, as a context for studying ways to manipulate data frames and review the data science process again, emphasizing how the stages interact and aren't always discrete. While cleaning data, we used skills visualizations and summary statistics to examine progress. This allowed for repetition and reinforcement. Databases served as context to discuss constructing datasets from pieces and a chance to reinforce concepts from accessing data.

In weeks 6-11, new machine learning models served as context to motivate different aspects of evaluation and modeling as shown in Table 2. We used the Sci-kit Learn API to take a model-centric approach, while sticking with the programming focus of the course (Buitinck et al., 2013). Sci-kit Learn provides a large number of typical models with a consistent API, this made it easy for students to try out new models independently to extend what was taught. We used object inspection in Python to examine the attributes of the estimator objects to learn about the model parameters and the built in Jupyter help to learn about the hyperparameters. Each mode motivated a new data science concept or skill: naive bays classifiers introduced the concept of classification and classifier performance; decision trees motivated cross validation; Support Vector Machines motivated parameter tuning; Knearest neighbors clustering and regularized regression motivated model comparison. This sequencing allowed for covering a variety of models and a straightforward path to reinforcing the core concepts and providing multiple opportunities to practice each skill. Models were presented with usage heuristics but without the algorithmic details of the `fit` methods.

The final three weeks covered nontabular data through case studies that allowed for reinforcement of many different skills and centered students interests. Nontabular data was framed as an extension of what was covered previously: we can represent data in tabular forms and then apply what we know. There was greater interest in natural language processing than in images, so we spent two weeks on text representations and only one week with images. After focusing on various representations of text, deep learning was presented as a complex model that can do both the representation and classification at once. This framing meant the last few weeks were another opportunity to reinforce everything covered to this point, giving students more opportunities to earn missed achievements if they were behind while still creating opportunities for students who were up to date to extend what they hard learned. Fireside chat style interviews with practicing data scientists, introduced helped students see that what we had covered in class directly connected to what data science is like in industry.

## 4. Class Sessions

During class time, I delivered instruction via participatory live coding, where the instructor types and explains code in real time and students follow along, typing the same code, getting practice in real time (Word et al., 2021; Nederbragt et al., 2020). Participatory live coding models realistic programming, students observe the instructor make mistakes, get errors, and debug them in real time. Error messages are difficult to parse for novices, so seeing the instructor parse and resolve them helps reach proficiency in this much faster than relying on internet searches alone. Additionally, debugging a model that does not perform as expected is an even more complicated process, but with this model, we can see this in real time[3]. With the aid of Jupyter Notebooks, students have a copy of the code produced in class, with their own notes. In the Carpentries, where this model of teaching was popularized, the audience is novices, who are coming to programming as a supplemental skill to support their research. In that context, the learners need a minimal mental model of how the code works to move into a competent practitioner role, these learners have good knowledge of what data analyses they wish to do and attend the workshop to learn to code as a tool. This course is a 300 level elective for computer science and data science majors; these students come to the class with significant prior experience in programming and need more depth in their programming, but with less experience in producing knowledge from data.

These differences require adaptations to the practice. First, the more advanced material and short sessions(50 minutes, 3 times weekly vs 2 day bootcamp) make some necessary code excerpts prohibitively long to type live. To accommodate, we used IPython load magic with a short url that pointed to the markdown download page for a

---

[3]e.g. while covering Decision trees

*Table 1.* Select Achievement Definitions. There are three achievements for each of 15 skills, describing a progression of learning for that skill. The keyword for each skill is a shorthand that was used throughout the course: from the schedule, to assignment text, and the gradebook. A full listing can be found on the syllabus[2]

| keyword | skill | Level 1 | Level 2 | Level 3 |
|---|---|---|---|---|
| **access** | Access data in multiple formats | Load data from at least one format; Identify the most common data formats | Load data for processing from the most common formats; Compare and contrast most common formats | Access data from uncommon formats and identify best practices for formats in different contexts |
| **visualize** | Visualize data | identify plot types, generate basic plots from Pandas | Generate multiple plot types with complete labeling with Pandas | generate and customize complex plots with plotting libraries |
| **prepare** | prepare data for analysis | identify if data is or is not ready for analysis, potential problems with data | apply data reshaping, cleaning, and filtering as directed | apply data reshaping, cleaning, and filtering manipulations reliably and correctly by assessing data as received |
| **classification** | Apply classification | Describe what classification is | Apply a prescribed classification model to a dataset | Select and apply appropriate classification models to different datasets |
| **compare** | compare models | Qualitatively compare model classes | Compare model classes specifically; compare performance of fit models | Evaluate tradeoffs between different model comparison types |
| **workflow** | Use standard tools to solve data science problems | Solve well structured problems with a single tool pipeline | Plan and execute solutions to fully specified problems; apply new features of standard tools | Scope, choose appropriate tools and solve open-ended data science problems; compare common tools |

*Table 2.* Course Schedule with skills emphasized each week. Skills are defined in Table 1 and linked by the keyword column

| Week | Topics | Skills |
|---|---|---|
| 1 | Overview, Python Review | python, process |
| 2 | Loading data | access, prepare, summarize |
| 3 | Exploratory Data Analysis | summarize, visualize |
| 4 | Data Cleaning | prepare, summarize, visualize |
| 5 | Databases & Merges | access, construct, summarize |
| 6 | Naive Bayes Classification | classification, evaluate |
| 7 | decision trees, cross validation | classification, evaluate |
| 8 | Linear Regression | regression, evaluate |
| 9 | Kmeans Clustering | clustering, evaluate |
| 10 | SVM, parameter tuning | optimize, evaluate, clustering |
| 11 | KNN, LASSO | compare, clustering, regression |
| 12 | Text Analysis | unstructured |
| 13 | Topic Modeling | unstructured, workflow |
| 14 | Deep Learning | workflow, compare |

HackMd[4] pad to import content to and editable notebook cell: `%load http://drsmb.co/310`. This method was most often used for import statements in the first cell of each class. The HackMD is editable in real time while maintaining a consistent url which allowed the instructor to add text there on the fly and share it with students immediately. Second, we used code inspection tools to examine data structures and class objects as a visual for conceptual discussions. For example, we printed out the object using `__dict__` attribute to see how the estimator object changed before and after fitting. We also made extensive use of built in Jupyter help views to consider parameters of methods before calling them. This both gave a visual to complement explanation and modeled for students where they could get help while working independently. This is important to model because after introductory courses that do everything from scratch in teaching -specific development environments many students come to this course unfamiliar with using documentation.

In class assessment occurred in Prismia chat[5], which provides a chat-like interface for students and allows the in-

structional team to see all student responses at once, group them, and reply individually or group-wise. Many questions were multiple choice questions designed to probe specific misconceptions, though some were open ended programming questions, where students submitted code to the chat. This served as formative assessment to reinforce concepts for students in real time and as way for the instructor to monitor progress.

At the end of class, students were able to submit additional questions through an Exit Ticket. Answers to those questions were appended to the instructor notebook prior to posting them online a course Jupyter Book. The instructor notes were also annotated with resources, written explanations, and extra practice exercises using Jupyter Book special content blocks after converting to Myst Markdown.

## 5. Assessment

In order to align assessment to the assumed model of skill acquisition, the course adopted a hybrid competency-specification based grading scheme. This grading scheme allowed specification grading of each activity, meaning that the instructor and teaching assistant did not have to calculate partial credit for assignments and that students had multiple chances to demonstrate each competency in the course. Specification grading involves defining a set of criteria, the specifications, for an assignment and assessing on a binary: the specifications are met or not(Nilson, 2015). In this case the specifications were the achievement definitions, crucially this allows for some mistakes to be made if the understanding is demonstrated. Standards based grading focuses on a student's best work rather than averaging all attempts and improves outcomes for marginalized students especiallyVerschelden, 2017, p64. Competency based grading allows students to work through material at their own pace and typically allows for resubmits on assignments. In this course, the grade was based on accumulated

---

[4]https://hackmd.io/
[5]https://prismia.chat/

achievements and there were multiple opportunities to earn each achievement, through the design of assignments, rather than resubmits. This repetitive structure in assignments also helped students see the material as connected as they were required to continue using skills as the semester proceeded.

Students had at least two opportunities to earn each of the 45 (15*3) achievements. Level 1 achievements could be earned on any type of activity: in class, assignments, or portfolio checks. Level 2 could be earned only on assignments and portfolios. Level 3 achievements could only be earned on portfolio submissions. Each skill was addressed in at least 3 class sessions, at least 2 weekly assignments and at least 2 portfolio submissions[6]. The communication learning outcome was built into all assignments and portfolios through the requirement to explain code and interpret results, using markdown cells in submitted Jupyter Notebooks.

Assignments were guided data analyses. Each allowed students to practice with new concepts and skills within a direct guidance. Each submitted assignment was graded on specification for level 2 achievement in each relevant skill independently; a student could earn a level 2 achievement for one skill, but not another in a given assignment. If the submission did not meet the specification for level 2, it was evaluated for meeting level 1. For example, correctly using and interpretting summary statistics, and choosing the right plot type failing to generate the plots in A3 would earn level 2 for summarize, but level 1 for visualize. Students submitted assignments as Jupyter notebooks to a GitHub repository created with GitHub Classroom from a template repository a GitHub Action to convert submitted notebooks files to Myst markdown with Jupytext(Team, 2020). The markdown format facilitated providing inline feedback through the Feedback Pull Request automatically created with GitHub Classroom by making a human readable file (Gennarelli, 2017; Team, 2020). This gave student explicit feedback about how to improve on future assignments aimed at encouraging students to continually improve(Verschelden, 2017). Because achievements were evaluated independently, students could skip portions of the assignment that assessed achievements they had already earned. For example, later assignments included suggestions for extra plots or modifications to the figures to include in order to earn level 2 for visualization. A student could also submit an empty repository indicating that they were choosing to attempt the relevant achievements through a different assignment.

Portfolios were a chance for students to demonstrate deeper understanding by building a large Jupyter Book in a single GitHub Repository over the course of the semester. Students wrote an introduction describing what achievements they were attempting to earn and where in their portfolio each was addressed. Portfolios were graded on specifica-

---

[6]full allocation on the syllabus

tion for the achievements the students identified. Students were provided with prompts to guide their inquiries to earn level three and the option to revise a previously submitted assignment to earn missed level two achievements. The open-ended prompts included both reflection-centered and analysis types and students were encouraged to propose alternative, creative options as well. To earn achievements for an assignment revision the student had to submit a more reflective notebook than was required the first time, describing where they were stuck or did not understand, addressing feedback they received, comparing their solution to the correct one if appropriate, and explaining the correct answer.

In the end, accumulated achievements were converted to a letter grade with a series of minimum thresholds shown in 3: to earn a C students had to accumulate all level 1 achievements; a B required all level 2 achievements; and an A required all level 3 achievements.

*Table 3.* Minimum Achievements required for each letter grade. For example, earning all level 1 achievements, 13 level 2 achievements and 6 level 3 achievements, would result in a B-.

| letter grade | Level 3 | Level 2 | Level 1 |
|:---:|:---:|:---:|:---:|
| A | 15 | 15 | 15 |
| A- | 10 | 15 | 15 |
| B+ | 5 | 15 | 15 |
| B | 0 | 15 | 15 |
| B- | 0 | 10 | 15 |
| C+ | 0 | 5 | 15 |
| C | 0 | 0 | 15 |
| C- | 0 | 0 | 10 |
| D+ | 0 | 0 | 5 |
| D | 0 | 0 | 3 |

# 6. Conclusion

This paper described Programming for Data Science, a programming focused introduction to data science with learning centered assessment. The design of the course was centered on student learning, but this organization also provides key advantages for the instructor. Grading without assigning partial credit is more streamlined; giving inline feedback on how students can improve their code, either to meet the specification, or be a better coworker is more enjoyable than taking points off. Having the clear learning outcomes that needed to be met with each activity that were written before the start of the semester made the ongoing prep lighter. Writing code live gives the freedom to adapt to student questions on the fly and means the advance preparation is only notes that will not be shared directly. Students appreciated that class went slow enough that they were able to really keep up with what was going on and because the live coding is active, the slower pace does not disappoint the more advanced students. Students shared later that they were able to apply concepts from class to do independent projects and use more complex models not covered because the programming patterns held the same.

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
