# OpenReview forum: "Participatory Live Coding and Learning-Centered Assessment in Programming for Data Science"
_ecmlpkdd.org/ECMLPKDD/2021/Workshop/TeachML — TeachML 2021_

### Official Review · Reviewer_NmZe · 2021-07-14
**Interesting report, the advantage of the presented method and the reason for wide coverage of ML methods in an entry-level course could have been explained in more detail**

**Rating:** 6
**Confidence:** 5

**Review:**

The paper meets the submission requirements. Apart from small typographical mistakes and slightly unclear formulations in some places, the writing style is good.

The authors report on a course called "Programming for Data Science". They state that they switched from the course using "slides and pre-filled Jupyter Notebooks followed by in class group work" to "participatory live coding", where students code along with an instructor. The authors could have elaborated more clearly on the advantages of the method. They state that "Participatory live coding models realistic programming, students observe the instructor make mistakes, get errors, and debug them in real time.". The authors could have explained why e.g. student group work, maybe under mentor supervision, wouldn't have the same effect. The reviewer has also taught courses using live-coding and gathered student feedback suggesting that coding along with an instructor while trying to understand the subject matter itself can be stressful for learners. It would be interesting to hear whether the authors have observed that as well.

The authors state that statistics or math skills are not a course prerequisite. Also they state that "For some students, this will be their only course exposure to machine learning concepts prior to graduation, so their motivations are largely practical.". Additionally, the course itself is said to be a prerequisite for a later Machine Learning course. However, the course already touches on quite a number of basic and at least intermediate ML methods (clustering, regression) and workflows (data loading, data cleaning, optimize model parameters, model performance evaluation, cross validation), covering large parts of scikit-learn's functionality. One could argue that this way of teaching bears the risk of students having some experience with an ML framework without solid understanding of the methods implemented therein. The authors could have given an explanation as to why so many ML topics are covered in what is presented to be an entry-level course, as well as in what depth topics are taught, especially comparing to the later Machine Learning course. Also, the authors could have reported to what extent students have, in the author's
experience, gained solid working knowledge of the ML methods studied such that they are able to confidently apply them, i.e. knowledge that goes beyond that which can be acquired by self-study of, say, scikit-learn's User Guide.


Additional comments:

L084: reference "2", should probably read "Tab. 2"

---

### Official Review · Reviewer_Hae2 · 2021-07-24
**A bit more explanation with perhaps a bit less detail here and there would strengthen this**

**Rating:** 7
**Confidence:** 3

**Review:**

A fundamentally interesting and detailed examination of a chance to a student-centered approach to teaching the basics of data science. I did expect a bit of an exploration of why the elements in the course and their sequence and how the tight schedule and the student-centered approach affects/affected what gets taught. E.g., a week for text analysis, unless structured in a way that I have failed to imagine, will likely result in students coming away with "it's complicated." And that may very well be the point. If it is, I think the author(s) is/are better off simply claiming it and making a case, however briefly, about the merits, and limitations, of doing so.

Some of the other confusions I had, I suspect, are a function of the brevity of the format. There's a lot to pack into very little space. Tables are seemingly ways to cram more in, and while I was fascinated by Table 1, I don't know that an abridged version with a bit more explanation might not have served better. (As the author(s) can probably discern, I look forward to an expanded version of this paper at some point.)

Finally, I appreciated the moment of intellectual history and contextualization that occurs on the second page and wish the paper had perhaps a slightly more extensive bibliography.

---

### Decision · Program_Chairs · 2021-07-24

**Decision:**

Accept

**Comment:**

Congratulations! The reviewers agree that this paper should be accepted.

Camera-ready version is due August 18, 2021. As you prepare the camera ready version, please take the reviewers comments into consideration.

We look forward to your participation at the workshop on September 13, 2021. We invite you also to join us for the satellite event on September 08, 2021. Schedules for both the workshop and the satellite event will be forthcoming.